# Removal of Anionic and Cationic Dyes Present in Solution Using Biomass of *Eichhornia crassipes* as Bioadsorbent

**DOI:** 10.3390/molecules27196442

**Published:** 2022-09-29

**Authors:** Eunice López-Ahumada, Mercedes Salazar-Hernández, Alfonso Talavera-López, O. J. Solis-Marcial, Rosa Hernández-Soto, Jose P. Ruelas-Leyva, José Alfredo Hernández

**Affiliations:** 1UPIIG, del Instituto Politécnico Nacional, Guanajuato 36275, Mexico; 2Departamento de Ingeniería en Minas, Metalurgia y Geología, División de Ingenierías, Universidad de Guanajuato, Guanajuato 36025, Mexico; 3Unidad Académica de Ciencias Químicas, Universidad de Zacatecas, Campus UAZ Siglo XXI, Zacatecas 98160, Mexico; 4UPIIZ, del Instituto Politécnico Nacional, Zacatecas 98160, Mexico; 5Facultad de Ciencias Químico-Biológicas, Universidad Autónoma de Sinaloa, Culiacán 73114, Mexico

**Keywords:** dyes, heterogeneity, active site, biomaterials, red phenol, gentian violet, red phenol

## Abstract

The discharge of large amounts of effluents contaminated with gentian violet (GV) and phenol red (PR) threatens aquatic flora and fauna as well as human health, which is why these effluents must be treated before being discarded. This study seeks the removal of dyes, using water lily (*Eichhornia crassipes*) as an adsorbent with different pretreatments. PR and GV were analyzed by a UV-visible spectrophotometer. Equilibrium experimental data showed that Freundlich is the best model to fit PR and SIPS for GV, showing that the adsorption process for both dyes was heterogeneous, favorable, chemical (for GV), and physical (for PR). The thermodynamic analysis for the adsorption process of both dyes depends directly on the increase in temperature and is carried out spontaneously. The Pseudo first Order (PFO) kinetic model for GV and PR is the best fit for the dyes having an adsorption capacity of 91 and 198 mg/g, respectively. The characterization of the materials demonstrated significant changes in the bands of lignin, cellulose, and hemicellulose, which indicates that the functional groups could participate in the capture of the dyes together with the electrostatic forces of the medium, from which it be concluded that the adsorption process is carried out by several mechanisms.

## 1. Introduction

Waste from the textile, leather, plastic, printing, and food industries, among others, severely affects the world’s water resources, since huge amounts of water are consumed producing large volumes of liquid waste considered highly toxic, teratogenic, and carcinogenic [1,2,3,4,5]. Because they contain dyes for the aquatic world, due to their dark color they block sunlight, which causes the photosynthesis process that is essential for aquatic life, and their presence in aquifers can induce dyes to enter the chain food [3,4,5,6,7,8]. For these adverse effects on public health, causing mutation-related, teratogenic, and carcinogenic diseases, dye removal has attracted great attention [1,2,3,4,5,6,7,8,9,10,11]. Among these dyes are gentian violet (GV) a synthetic cationic dye also known as gentian violet (Figure 1a) [8]. This dye is widely used in the textile and clothing industry to dye cotton, wool, silk, and nylon, in the manufacture of printing, inks, and as a biological colorant used to classify bacteria, dermatological agents in veterinary medicine, and human skin disinfection [12,13,14,15]. However, GV is toxic and can be absorbed through the skin causing in extreme cases kidney failure, blindness permanent, and cancer and with 1 ppm it can cause very severe damage to bodies of water since it can cause a significant reduction in the presence of sunlight, inhibiting photosynthesis and producing microtoxicity in aquatic flora and fauna due to its easy adsorption on the skin of fish reducing its metabolism and facilitating the formation of tumors in some species [5,6,7,8,9,10,11,12,13,14,15]. Phenol red (PR) is presented as a red and odorless powder or crystals whose formula and chemical structure are shown in Figure 1b. It is used as a fabric dye [15,16,17]. PR is widely used in in vitro toxicological studies and culture media it is used as an indicator of pH change (between 6.4 and 8.2) [16,17]. It has another use because it has a structural resemblance to some estrogens that do not have natural corticosteroids, which allows it to act as a weak stimulator of their receptors, among other uses, and can affect the respiratory system and human skin, although it is not considered carcinogenic in low concentrations [16,17].

Due to the great concern that these colorants represent, various conventional physical-chemical processes have been used (coagulation, photocatalytic, ion exchange, flocculation, adsorption, membrane filtration, and irradiation), to remove them from water obtaining a good decolorization efficiency [18,19]. Nevertheless, these proceeds have two main limitations: production of by-products and the high cost of maintenance [2,5,18]. On the other hand, biological treatment is not effective for this type of contaminant due to the low biodegradation of dyes [2,13,19,20,21,22]. Adsorption has shown to have some advantages over the mentioned methods due to its high efficiency in the removal of dyes from wastewater including dilute solutions, and the absence of sediment produced in its operation [4,12,19,23,24,25,26]. Different adsorbents have been synthesized for the removal of dyes from clays such as SBA-15, Bentonite, and other materials [2,4,17,27,28,29], up to the manufacture of activated carbon from different sources, however, there are large losses and high operating costs due to the possibility of pore blockage, hydroscoping, and incineration when using high temperatures [5,19,22,24,27,28,29]. These drawbacks have encouraged several researchers to study other cheap, abundant materials that have similar efficiency to materials synthesized by chemical processes, therefore the use of agro-industrial residues was considered, which are cheap and, easily available materials that only need a simple pretreatment and are materials with properties that can be exploited in another industry as well as being good candidates for the adsorption of various contaminants [8,12,14,19,22,26,27,28,29,30,31,32,33,34]. These materials are mainly constituted by lignocellulosic groups, peels of various fruits and vegetables furthermore, these materials have been used as adsorbents of various dyes including PR and GV with good results [4,5,11,12,13,18,22,23,24,25,26,33,35,36].

In recent years, a technique has been used that aims to protect the environment of water systems using treatment systems with aquatic plants, making this technique a phytoremediation process for the removal of heavy metals and dyes from water [1,36]. Some freshwater macrophytes used as bioadsorbents are *Potamogeton lucens*, *Salvinia hergozi*, *Myriophyllum spicatum*, *Nymphaea alba*, *Cabomba* sp. and *Cratophyllum demersum* which have been found to have several mechanisms for the elimination of contaminants such as sorption/precipitation on the cell surface, intracellular accumulation and extracellular accumulation/precipitation [3,19,27,28,37,38,39,40,41,42,43,44]. Among these aquatic plants, water lily has also been considered (*Eichhornia crassipes*), which has a large amount of lignin, cellulose and hemicellulose in its biomass (>60%), however, it is considered a harmful plant because it invades the inhabitant of lakes, lagoons where its proliferation is very accelerated, this causes oxygen to be consumed by this plant killing fish and prevents sunlight from reaching aquatic plants and is the main habitat of mosquitoes [19,22,27,31,35,38,39]. But due to its tolerance and ability to adsorb high concentrations of pollutants and it is considered a promising material to be used in the bioremediation of wastewater, for which it has received great attention in recent years for its possible economic application and efficiency for the removal of dyes [18,35,36,37,38].

Based on this, in this work the adsorption capacity of water lily (WL) extracted from Yuriria lake in Guanajuato, Mexico treated with water (WLW) and NaOH (NWL) was studied to understand the effect of the modification of the surface of WL. The adsorption mechanism of the dyes as well as the influence of important parameters such as contact time, the amount of bioadsorbent, initial concentration of the dyes, the pH of the solution together with the participation of the functional groups present on the surface of the treated WL that is involved in the removal process for which different kinetic and isothermal models. Also, it was considered a study of the influence of external and/or internal mass transfer in the process of GV and PR removal from aqueous effluents.

## 2. Results and Discussion

### 2.1. Effect of the Initial Concentration of Dyes on the Percentage of Removal

In bioadsorption, some parameters play an important role during the removal of contaminants, and thus knowing the consequences that these parameters have during the global process is mandatory [36]. The analysis for dyes removal with different the initial PR and GV concentrations with the bioadsorbent with different treatments) can be seen in Figure 2. It was observed that the percentage of removal increases with the increase in the initial concentration for both treatments, having a greater efficiency with WLW (around 22%) compared with the bioadsorbent NWL for PR, and in the removal of GV, the same is presented trend having about 34% increase in the percentage of because having a greater number of molecules of the dyes in solution increases the interaction between them, causing the interaction between the dyes and the surface of the bioadsorbents to increase significantly, facilitating the removal of the dyes in solution. The bioadsorbent treated with water had a better percentage of removal when compared to the bioadsorbent treated with NaOH irrespective of the dye. Studies reported in the literature mention that there is a decrease in the adsorption of the dye when using modified lignocellulosic waste like “Quimbimbo” [11] and hydrophilic silica aerogel [2] as opposed to Bentonite [29], modified Cliboptilolite [23], banyan aerial roots [13], AlPO-34 [4] and the bottom ashes of the materials used in the soybean deoiling process [33] where the adsorption of GV and PR increases with increasing concentration of the dye. This increase is possibly favored by the increase in the number of collisions between the molecules, causing the adsorption process to be facilitated and therefore a significant decrease in the mass transfer between the liquid and solid phase.

The adsorption process has a direct relationship with the temperature, when the temperature increases, the percentage removal of the dyes increase. It was observed that the maximum value of percentage removal at 60 °C of PR was 98.8 and 85.6% in WLW and NWL, respectively, at a concentration of 12 ppm and for GV this maximum value was obtained with a concentration of 32 ppm of 92.45 and 58.5% in WLW and WL, respectively at 60 °C. This represents a loss of adsorption capacity due to the treatment of around 13.4% for PR adsorption and 36.7% for GV, however, the maximum removal loss occurs at 30 °C for PR (27.5%) and 45 °C for GV (39.75%). Some studies indicate that the loss in adsorption efficiency of contaminants is directly related to the pretreatment of the bioadsorbent, although other studies show a similar trend to our results [39,40,41,42,43,44,45]. The reduction of initially available sites on the surface of the adsorbent, to capture the ions of both dyes, could explain the loss in adsorption. The analysis of this parameter is very important because all adsorption processes involve mass transfer, which is related to this and other parameters.

### 2.2. Equilibrium Dye Adsorption

The analysis of the different adsorption isotherm models (Table 1) allows obtaining information on the process type carried out during the removal of pollutants.

It is shown below, in Figure 3 and Figure 4, the different adjustments made to the isotherms obtained for gentian violet (GV) and phenol red (PR), for the different treatments and temperatures, observing a decrease with the adsorption of both dyes with temperature. It was observed that the maximum adsorption at 60 °C for GV using treated water lily (WLW) and (NLW) was 63.4 and 39 mg/g, respectively. This reveals that the effect that the treatment has on the bioadsorbent represents a decrease of 40 to 45% in dye removal, indicating that the modification in the surface of the biomaterial directly influences the adsorption capacity.

In the case of PR, the maximum adsorption obtained was 25.3 mg/g for WLW and 20.2 mg/g for NWL at 60 °C, this implies that a loss of the removal is around 20%, although there is a smaller decrease in the adsorption for this dye when compared to GV with the same treatment for the material. In addition, it can be observed for both dyes that the adsorption capacity increases with the temperature value, showing the nature of the adsorption process.

Table 2 shows the results of the different parameters of each isotherm model, considering the normalized standard deviation (Δq%) as well as the deterministic criterion (R^2^) to choose the best fit. For the case of PR in both treatments, it was found that the dye removal process is carried out with a heterogeneous surface using more than one layer (Freundlich) having physical adsorption (n > 1). Additionally, the adsorption energy determined with the DR model decreased with the increase in temperature, which implies a favorable effect for the dye adsorption with the increase in temperature and explains observations in Figure 3 and Figure 4. The Langmuir model suggests favorable adsorption (0 < R_L_ < 1). Even though both treatments present the same adsorption process (Physical), for the adsorbent treated with water (WLW), a higher adsorption capacity and removal percentage (84.4%) was observed compared to the adsorbent treated with hydroxide (67.4% for NWL).

The results in the adsorption of PR in different adsorbents were adequately adjusted to the Langmuir model using Bentonite obtaining an adsorption capacity of 166.7 mg/g [17] and activated carbon derived from rice husk [5] having 6.77 mg/g with 90% while using modified Clinoptilolite the best model was Freundlich with an adsorption capacity of 0.35 mg/g [23], which shows that PR removal using treated WL is a good candidate to eliminate this dye from the effluents of the industries that produce this waste since our results show practically the same removal percentage and an adsorption capacity within the values reported in the literature.

Table 3 contains the different parameters of the GV adsorption isotherm, these values indicate that the best fit is presented by the Sips model, showing the heterogeneous nature of the active site surface, revealing that the dye is removed by a chemical process (n_s_ < 1). This is confirmed by the value of n of the Freundlich model (less than 1) and Langmuir indicates favorable adsorption (0 < R_L_ < 1). In addition, the removal energy determined with the DR model showed that it increases with the increase in temperature, confirming the observations with the adsorption of PR, and suggesting an increment in dye adsorption with the increase in temperature. Although both treatments present the same adsorption process (chemical) a higher removal percentage (98%) can be observed for the adsorbent treatment with water, meanwhile, the removal percentage for the adsorbent treatment with hydroxide was lower (71%). Several models have been reported in the literature to explain the adsorption process with different adsorbents, such as Langmuir, with reported values of adsorption capacity between 18.50 and 1273.11 g/mg [4,8,29,36]. Other works obtained an adsorption capacity of 3.54, 32.33, 413 and 456.64 mg/g adjusting their experimental data with the Boer-Zwikker [2], Weber-Morris [5], Redlich-Peterson [11], and Temkin [13] models, respectively. On the other hand, in the adsorption of GV using *Diplazium esculentum*, it obtained an adsorption capacity of 350.64 mg/g with a removal of 96% and Sips as the best model [12], demonstrating similarity in our results and therefore the viability of using treated WL as a viable candidate for the complete removal of GV in effluents.

### 2.3. Thermodynamic Analysis of the Adsorption Process

To gain insights into the nature of the adsorption process of the dyes with WL as adsorbent, the thermodynamic parameters were analyzed and are shown in Table 4. In all cases the Gibbs energy was negative (ΔG < 0), attributed to spontaneous process. It also confirms the observations in Figure 3 and Figure 4, the adsorption capacity increases when the temperature increases, supported by the positive values of enthalpy (ΔH > 0). The forces present on the surface of the adsorbent participate significantly in the process along with the interaction in the solid-liquid. This behavior can also be determined independently of dye adsorption and treatment. This behavior has been reported in other studies with different adsorbents used for the removal of PR and GV [12,13,17,20,29,33].

### 2.4. Contact Time and Concentration of the Bioadsorbent

Another significant parameter for accelerated removal of these dyes with optimal efficiency and practical application is the contact time of the adsorption process, a key parameter to establish the last stage of the adsorption kinetics in heterogeneous systems, which is the equilibrium that is found given by the accumulation or saturation of the sites available for pollutants [40,41,42,43]. Along with the other two stages: the first is rapid initial adsorption, since there is a great availability of sites for removal, together with an instantaneous diffusion from the core of the solution to the surface of the biomaterial, this step is followed by progressive adsorption until equilibrium is reached, where there may be problems of external or internal mass transfer [36,39,41]. Figure 5 and Figure 6 show the contact time of PR and GV with the treated biomaterials where it is observed rapid adsorption for both dyes until there were no significant changes in adsorption capacity over time regardless of the treatment and temperature, thus, reaching equilibrium. At this point, the optimal time for the maximum adsorption capacity was ~3 h and ~2 h for PR and GV, respectively. Additionally, it could be inferred that the sites on the adsorbent surface already do not allow the transfer of dyes within the solution [2,13,22,33].

Figure 6 shows the effect of the adsorbent concentration (C_ads_) for both dyes with the different treatments. This analysis was carried out using an adsorbent concentration range of 0.1 to 0.5 g/L with a concentration of 10 and 20 ppm of PR and GV, respectively. In this figure, it was noted that a decrease in the adsorption capacity and the increase in C_ads_ could be due to blockage of the active sites found on the surface of the biomaterial, in addition to having a relationship indirectly proportional to the adsorption capacity. It was also observed that there is an increase in the percentage of removal with the increase in the amount of mass of the adsorbent, since the adsorption of both dyes increases due to the increase in the area available for the exchange process between the solution and the surface until reaching the equilibrium. This type of behavior has been reported in other studies with different adsorbents [2,13,22,42,43,44,45,46,47,48,49]. This analysis allows us to understand and know the limitations due to mass transfer, so this particularly important parameter must be considered [2,13,23,33].

### 2.5. Adsorption Kinetics

Understanding the behavior of the dynamic adsorption process of the dyes in the biomaterials by adjusting the experimental data with the kinetic models (Table 5) allows us to know the control mechanism during the removal of dyes. To choose the best model used two criteria, one of them is the deterministic coefficient (R^2^) and the other one is the normalized standard deviation (Δq%). These parameters along with additional parameters belonging to each model are presented in Table 6 for PR adsorption with WLW and Table 7 for PR adsorption using NWL. For the PR with WLW, the best fit is represented by the PFO model, pointing out the need for an active site on the surface of the bioadsorbent for each PR molecule since the adsorption sites are widely available for elimination in the solution. In addition, the ID and ED models are not the most adequate, possibly explained by the absence of mass transfer limitations. The adsorption capacity at 30, 45, and 60 °C was 64.5, 69.8 and 91 mg/g, respectively, which represents an increase of 29% with increasing temperature.

To know the effect on the surface of the modified WL using NaOH, the adjustment of the kinetic data of the PR adsorption was carried out (Table 7). For this bioadsorbent, the same adsorption mechanisms were found in the removal of the dye with WLW, that is, that in the removal process a surface active site is needed for each molecule of the dye and there are no limitations due to internal and external mass transfer. The maximum adsorption capacity obtained was 71.62 mg/g at 60 °C and as the temperature decreases, the capacity decreases, being 64.54 and 55.41 mg/g at 45 and 30 °C, respectively. Comparing the results obtained for both adsorbents in the adsorption of PR, it is concluded that modifying the WL surface with NaOH caused a loss of about 22% of the maximum adsorption capacity as an effect of the treatment. However, the nature of the adsorption process did not change irrespective of treatments, since in both cases the adsorption capacity increases with increasing temperature, confirming what was determined with the thermodynamic analysis of the process.

In the literature, there are reports using the PSO model [17,23] and obtaining a maximum capacity between 0.5 and 142 mg/g, values consistent with the result here presented, although there is a discrepancy. With the model obtained in this work, this could be explained by the use of different pretreatments and the nature of the dye, allowing greater interaction between the surface of the bioadsorbent and the solution, since if it is positive, the electrostatic forces caused between the two, on the other hand, if the surface is negatively charged, cause repulsion forces between the dye and the surface as in our case of the treatment carried out on NWL.

The kinetic parameters of the elimination of GV using WLW and NWL are shown in Table 8 and Table 9, respectively. These values reveal that the adsorption of the dye is carried out on the surface of the biomaterial through the active sites in a ratio of one dye molecule per active site (PFO), in addition to not having problems due to mass transport during the process regardless of the treatment used in WL, allowing other factors to intervene significantly during the adsorption process. In the case of WLW, the maximum adsorption capacity of GV was found at 60 °C (198.4 mg/g), and decreases with decreasing temperature, since at 45 and 30 °C it has an adsorption capacity of 181 and 172 mg/g, respectively, representing a decrease of about 13%. For the adsorption of GV using NWL, the adsorption was 168.6, 159 and 149.7 mg/g at 60, 45 and 30 °C, respectively. The modification of the surface treated with NaOH directly affects the adsorption capacity by 15%, in addition, the nature of the adsorption process is endothermic, by the thermodynamic study.

In the literature has been reported GV adsorption that the best model is the PSO with adsorption capacity values of 90.9–1125 mg/g, employing mainly biochar, Bentonite, and modified cane bagasse, among others [5,8,13,17,36]. However, in a study conducted by Nayar et al. [10] the best model to explain their data was PFO with an adsorption capacity of 412.99 mg/g and modified lignocellulosic waste (Quimbombo) as bioadsorbent. There are some similarities in compositions (cellulose, hemicellulose, and lignin among other materials) between Quimbombo and WL, therefore, it is expected a similar behavior when as bioadsorbent of this dye. Also, because the surface of WL is negatively charged, it facilitates the removal process by electrostatic attraction due to the positive nature of the dye. In addition, based on the information obtained from the Avrami model, not only do these forces intervene in the removal of GV and PR, other mechanisms (physical, chemical, a combination of both, ion exchange, etc.) are involved. Table 10 shows the adsorption capacity of several bioadsorbents where it was observed that the results of our study have similar efficiency and resistance of the specific dyes, phenol red, and gentian violet.

This allows a good capacity of adsorption by using a cheap and easy treatment with biomass that represents an environmental and economic problem in lagoons, and rivers where it is found, but with adequate pretreatment, it becomes a potential candidate for the adsorption of dyes due to its high availability, versatility and capacity adsorption.

### 2.6. Characterization of Biomaterials

Figure 7 shows the SEM images of the biomaterials treated with water used in the adsorption of PR and GV, it can be seen that the surface presents anomalies along with cavities and porosity, these characteristics allow an excellent probability to have an excellent adsorption capacity of dyes on the adsorbent surface [27,32,43]. In addition, it was possible to observe on the surface of the bioadsorbent, particles of the dyes that were deposited and adsorbed within cavities or pores, demonstrating excellent properties on the surface of the material for trapping and retaining dyes.

Figure 8 shows the micrographs of the adsorbents treated with NaOH after the bioadsorption process of PR and GV. The micrographs contribute evidence of dye molecules adsorbed on the bioadsorbent surface. Moreover, some fibrous surfaces with fractures and fewer porous can be noticed when compared to WLW [27,32,43]. With these differences, the lower adsorption capacity is explained, as bearded out the kinetic and equilibrium results. These results for both dyes, here presented and discussed, are very promising and encouraging for dyes removal using Water Lily (*E. crassipes*) as a readily available and natural adsorbent [38,44].

The elemental analysis of WLW and NWL was conducted using X-ray dispersion spectroscopy (EDS), obtaining the results shown in Table 11 where it was possible to perceive a significant change in the C/O and Ca/Si ratio of both biomaterials before and after the dye adsorption process. The C/O ratio for WLW and WLN was 1.57 and 2.35, respectively, which indicates that both treatments affect differently the functional groups of cellulose, hemicellulose, lignin, etc., found on the surface of WL [45].

For the Ca/Si ratio of the same sample, it was 2.39 and 1.37, where it could be understood that the treatment with water does not significantly affect the ratios of Ca and Si ions compared to the treatment with NaOH where there is a difference of 42.68% concerning WLW. When studying biomaterials after the bioadsorption process, a decrease in these ratios was observed in both dyes. In the case of PR, there is a decrease in the C/O ratio present in WLW and NWL of 38.72 and 38.5%, respectively, while in the Ca/Si ratio it was 28.71 and 90.5%, respectively. This result shows that adsorption is not favored if the main participant functional groups contain Si and Ca on the surface. On the one hand, if the participation occurring mostly is from the groups containing C and O, there will be a better adsorption capacity for PR that in this case, everything indicates that it is with WLW. On the other hand, for GV adsorption the opposite occurs, since the decrease for the Ca/Si ratio present in WLW and NWL was 60.58 and 17.59%, respectively, and for the C/O ratio, it was 55.32 and 19.71%, respectively, which indicates that the cationic nature of the dye, together with these functional groups, are the ones in charge promoting better removal than in the case where WLW was used. The results allow us to mention that there could be more than one adsorption mechanism involved, such as (i) electrostatic forces, (ii) intervention of functional groups present on the surface, and (iii) ion exchange [45,46,47,48,49].

The diffractograms shown in Figure 9 contain the peaks at 14.7, 21.9, 24.3, 28.3, 40.5, 50, and 58.5° for WLW, with the first three peaks referring to amorphous cellulose, the following peak (28.3°) is the characteristic pattern of WL and the other peaks (40.5, 50, and 58.5°) are assigned to the presence of calcite in the adsorbent [31,33,47]. Comparing this pattern with NWL we realized that there are significant changes in the NWL sample where the intensity of the peaks at 14.7 and 80° decreases, this is probably to a change in the crystallinity of the sample or the lignin removal in WL. In addition, in the peaks of 14.7 and 21.9°, there are changes in the bandwidth which allows us to confirm that there are changes in the crystal lattice of WL [32,49].

After the adsorption of GV and PR with both bioadsorbents (Figure 9) it was possible to notice that all the peaks disappeared exception of the signals at 14.7 and 21.9°, this effect is assigned to crystallinity increment of the materials increases [34,48,49]. The lattice parameters were also shown in Table 12. There, the changes in the lattice parameter (a_0_) can be distinguished which suggests that crystallinity can intervene during the dye removal process.

The study carried out using ATR-FTIR in the different bioadsorbents is shown in Figure 10. The different bands observed represents the functional groups detected on the surface of WL. In this spectrum, the stretching vibration found at 3300 cm^−1^ is related to the OH group and the bonds corresponding to lignin, cellulose, and hemicellulose, while at the signal 2921 cm^−1^ it is assigned to the symmetric and asymmetric vibrations of the C-H bond of the methyl and methylene groups on the surface [38,48,49]. The band at 2859 cm^−1^ is assigned to the stretching of CH groups of lignin [45]. The bands at 1628 and 1536 cm^−1^ correspond to the stretching vibrations of the C=O carboxylic bond and lignin, respectively [32,38,48]. There is a peak at 1413 cm^−1^ corresponding to the vibration of aliphatic compounds (CH_2_ and CH_3_) and methoxy group (O-CH_3_) [45] whilst the peaks at 1320, 1240, and 1013 cm^−1^ are attributed to the symmetry vibration of the stretching of the COO- bond, to the presence of the C-H bond of the aromatic group and the stretching vibration of the C-OH bond of the groups’ alcoholics, lignin and carboxylic acids, respectively (40,60). In this spectrum, two shoulders are found at 1157 and 1044 cm^−1^, evidence of the stretching vibration in the C-O bond of the lignin structure and the vibration of the CO-R bonds of the alcohol groups, respectively [45,48]. The bands at 900 and 675 cm^−1^ are due to stretching of the C=O group and the β-glucosidic bonds of the cellulose, respectively [48,49]. About the WLW and NWL spectra, it was observed that all the bands, including the shoulders, are affected by the WL treatment, specifically, those related to lignin (3300, 1628, 1536, 1246 and 1013 cm^−1^) causing an increase in the degree biomaterial of crystallinity [49] and therefore the following order was obtained: NWL > WLW > WL, which confirms what was noted in XRD.

Figure 11 shows the spectra of WLW and NWL after the adsorption of PR and GV, where it was possible to distinguish that there are distinct functional groups present in the biomaterials. The peaks related to lignin, carboxylic groups, etc., were decreased in their intensity. However, this behavior is more noticeable for the adsorption carried out by WLW for both dyes (Figure 11a,c) contrary, for the removal carried out with NWL, this interaction between the functional groups on the surface of the biomaterials and the dye is less (Figure 11b,d) specifically with PR due to its anionic nature, causing repulsion forces between the surface and the dye. With WLW, which favors the adsorption of a cationic dye, attractive forces are produced that benefit its adsorption [34,38,48].

By determining the isoelectric point of the bioadsorbent (WLW, 6.5 and NWL, 4.74) and knowing the pH of PR and GV solutions (8.26 and 3.5, respectively), expected to have a negatively charged surface (pH_sol_ > pH_pzc_) for the adsorption of both dyes except for GV removal using WLW (pH_sol_ < pH_pzc_), where there is a positively charged surface [5,11,34,49]. The removal of PR with WLW is not complete because there are repulsive forces involved in the process. The same occurs with the removal of GV and employing the same bioadsorbent, where the participation of the functional groups on the surface of the material intervenes to a greater extent than the electrostatic forces, this behavior is also present in the adsorption of PR using NWL. Furthermore, using NWL there is greater participation of the electrostatic forces in the adsorption of GV compared to the functional groups present in the bioadsorbent, although the adsorption is not greater than that with WLW because the forces of attraction are weak compared to those produced with interaction with functional groups, thus confirming what was obtained in the kinetics, adsorption isotherm, and the ATR-FTIR analysis.

## 3. Materials and Methods

### 3.1. Reagents

All reagents used were analytical grade. Deionized water was used for the preparation of all the solutions. The dyes, gentian violet (GV, λ**_max_** = 585 nm) and phenol red (PR, λ**_max_** = 435 nm) of Meyer Chemistry, were used without any additional purification [2,17].

### 3.2. Preparation and Treatment of Water Lily (WL)

The water lily (WL) from Lake Yuriria, Guanajuato, Mexico was washed with running water at room temperature, dried in a forced convection oven (Shel Lab CE5F, Cornelius, OR, USA) at 80 °C for 24 h, and was crushed with an industrial blender (Tapisa Acero Inoxidable, City of Mexico, Mexico) until a fine powder was obtained (100 mesh), for subsequent pretreatment. The water lily was subjected to a treatment with deionized water (WLW) with a ratio of 30 g/L (P/V) at 75 °C and constant stirring for 1 h. Then the solution was filtered with a vacuum pump (Thomas 1CZC8, ON, Canada). This process was repeated until obtaining a crystalline filtrate. Dry the solid was dried at 85 °C overnight in a forced convection oven. Another treatment for WL was conducted with sodium hydroxide (NaOH), at 0.5 M (WLN) with a ratio of 30 g/L (P/V) and had an initial pH of 13.65, stirred for 2 h at 60 °C. Subsequently, it was filtered and the solid was mixed with a 1 M HCl solution (pH~1) while stirring for 1 h at room temperature. After this time the solid was filtered and washed with 10 times the amount of water used in the HCl solution. Finally, dry the solid was dried at 85 °C overnight in a forced convection oven. With this treatment, for every 100 g of initial plant, 81.5 g of a WL powder is obtained (18.5% loss weight) for $7 and $13 for the treatment with water and NaOH, respectively.

### 3.3. Equilibrium Dye Removal (Adsorption Isotherms)

For the equilibrium study of the dye adsorption, a mass ratio of adsorbent/volume of a solution of 0.5 g/L was used. The concentration varied from 0 to 40 ppm for GV and from 0 to 15 ppm for PR. The experiments were performed in a shaker (ZHWY-200D, Huizhou, China) at 200 rpm at 30, 45, and 60 °C until equilibrium was reached (~11 h of contact time). The samples were centrifuged (Generic 6-TRPR, Gosheim, Germany) at 6000 rpm for 10 min. UV-Vis spectrophotometry (VELAB VE-5000, City of Mexico, Mexico) was used to analyze the concentration of the different dyes present in the solution. The amount of dye removed by the adsorbent at equilibrium, q_e_, was obtained with the following expression [34,49]:(1)$$q_{e}=\frac{V\left(C_{0}-C_{e}\right)}{m}$$
where C_0_ and *C_e_* are the initial concentration and in equilibrium (mg/L), V is the volume of solution (L) and m is the mass of WL (g). The removal percentage, %R, was calculated as follows [34,49]:(2)$$\%R=\frac{\left(C_{0}-C\right)}{C_{0}}*100$$

The regression coefficient was calculated to evaluate the fit of each nonlinear model and the separation factor, R_L_, which allows for predicting the affinity between the adsorbent and adsorbate, using the following equation [34,49]:(3)$$R_{L}=\frac{1}{1+K_{L}C_{0}}$$
where K_L_ is the constant of the Langmuir model and C_0_ is the initial concentration of dyes. To understand the thermodynamics of the adsorption process, thermodynamic parameters such as apparent Gibbs free energy were determined.
(4)$$\Delta G=-\mathrm{RTln}\left(55.5K_{L}\right)$$
where K_L_ is the constant of the Langmuir (L/mol) model, R is the ideal gas constant and T is the absolute temperature (K).
(5)$$\ln\left(55.5K_{L}\right)=\frac{\Delta S}{R}-\frac{\Delta H}{\mathrm{RT}}$$

The values of ΔH and ΔS can be determined with the slope and sorted to the origin of the ΔG chart as a function of 1/T [34,49].

### 3.4. Batch Dye Removal (Adsorption Kinetics)

The adsorption kinetics was performed using GV solution at 20 ppm and PR solution at 10 ppm, the adsorbent concentration was varied from 0.1 to 0.5 g/L, agitated very 1.5 h and centrifuged at 6000 rpm for 10 min. These samples were analyzed to evaluate the concentration of the different dyes present in the solution by UV-Vis spectrophotometry. The amount of dye removed by the adsorbent, q, was obtained with the following expression [34,49]:(6)$$q=\frac{V\left(C_{0}-C\right)}{m}$$
where C_0_ and C are the initial and final concentration (mg/L), V is the volume of solution (L) and m is the mass of WL (g). In addition to using the coefficient of determination to compare the efficiency of the different kinetic and equilibrium models, the standard deviation, Δq%, was calculated [34,49]:(7)$$\Delta q\%=100*\sqrt{{\frac{\left(\frac{q_{\exp}-q_{\mathrm{cal}}}{q_{\exp}}\right)}{N-1}}^{2}}$$
where N is the number of data, q_exp_ and q_cal_ (mg/g) are the experimental and calculated values of the removed dyes, respectively.

### 3.5. Characterization of Bioadorbents

Attenuated Total Reflectance-Fourier transform spectroscopy (ATR-FTIR) analyses before and after adsorption of DNS were carried out over the wave number range of 4000–400 cm^−1^ using a Thermo Scientific Nicolet iS10 analyzer (Waltham, MA, USA), 32 scans were obtained with a resolution of 4 cm^−1^. X-ray diffraction patterns (XRD) were obtained in a diffractometer (Ultima IV Rigaku, Tokyo, Japan). To determine the isoelectric point, a sample of adsorbent in water with an initial ratio of 0.05 g was stirred at 200 rpm for 24 h in 50 mL to determine its pH with a potentiometer (Science Med SM-25CW, Dantali, India), 0.05 g was added every 24 h until the pH did not change [48]. The scanning electron microscopy images and the X-ray energy dispersion spectroscopy (SEM-EDS-EDX) were obtained in a JOEL spectrometer (6510 pus, Peabody, Boston, MA, USA). spectrometer (6510 pus).

## 4. Conclusions

In this study, it was established that *Eichhornia crassipes* (WL) modified with H_2_O and NaOH are good candidates to be used as bioadsorbents in the removal of phenol red (PR) and gentian violet (GV) in aqueous solutions. The adsorption of dyes was carried out on a heterogeneous surface using more than one monolayer removing around 85% of PR and 98% of GV from the solution using WLW. The removal of dyes decreases significantly when NWL was used resulting in dye adsorption of 67.4% for PR while GV was eliminated at 71%. There are different characteristics on the surface of the adsorbents due to the treatment used, however, have similar dynamic mechanisms were using one site per molecule. The adsorption capacity at 60 °C of 91 and 198.4 mg/g was obtained for PR and GV, respectively, and as the temperature increases the adsorption capacity increases. The characterization of both bioadsorbents before and after adsorption allowed us to establish a complete panorama of the dye removal process, showing that there are electrostatic forces, modification in the crystalline structure, ionic exchange caused by the participation of functional groups, and that the nature of the dye (anionic and cationic) can also be used together with the modification of the surface of the material considering the isoelectric point.

## Figures and Tables

**Figure 1 molecules-27-06442-f001:**
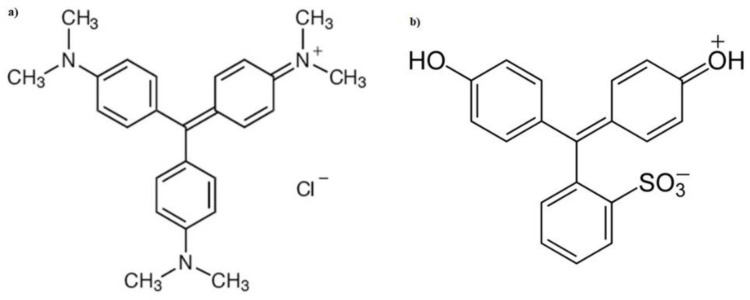
Chemical structure of: (**a**) Gentian Violet and (**b**) Phenol red [8,16].

**Figure 2 molecules-27-06442-f002:**
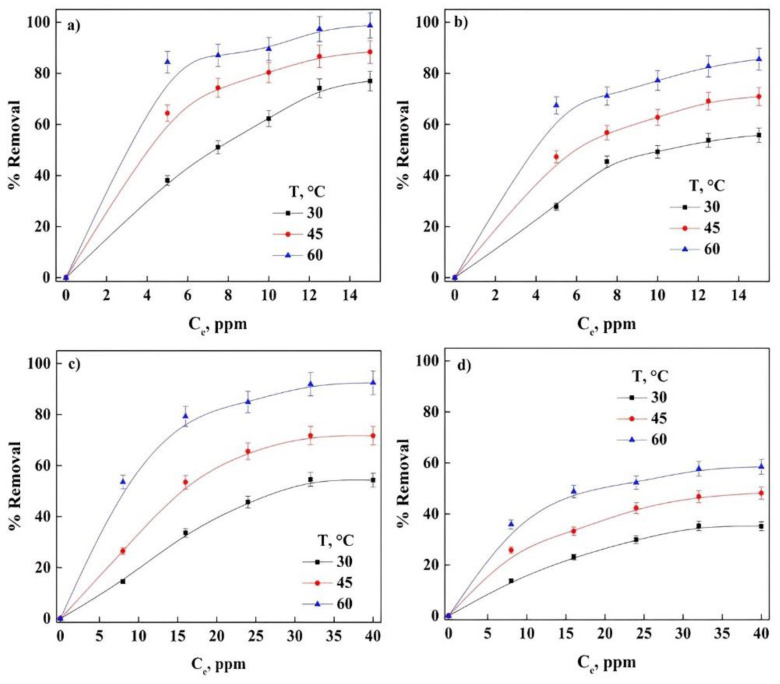
Effect of initial concentration of dyes: (**a**) WLW-PR, (**b**) NWL-PR, (**c**) WLW-GV and (**d**) NWL-GV.

**Figure 3 molecules-27-06442-f003:**
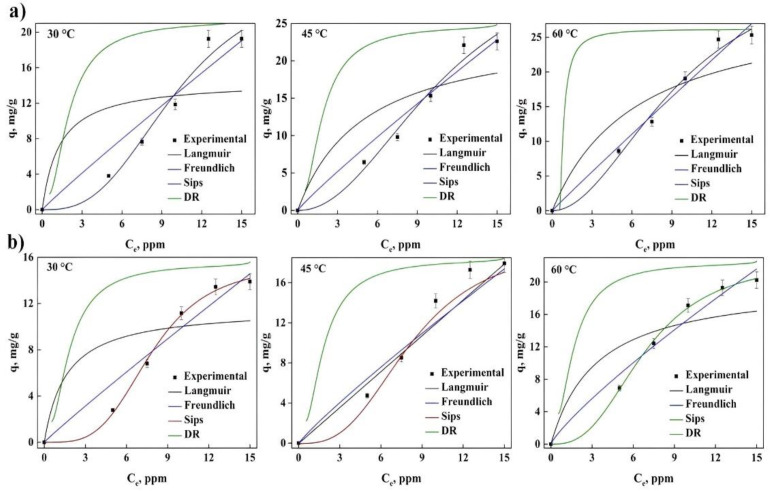
Fitting for the experimental data of PR adsorption at different temperatures employing different isotherm models: (**a**) WLW and (**b**) NWL.

**Figure 4 molecules-27-06442-f004:**
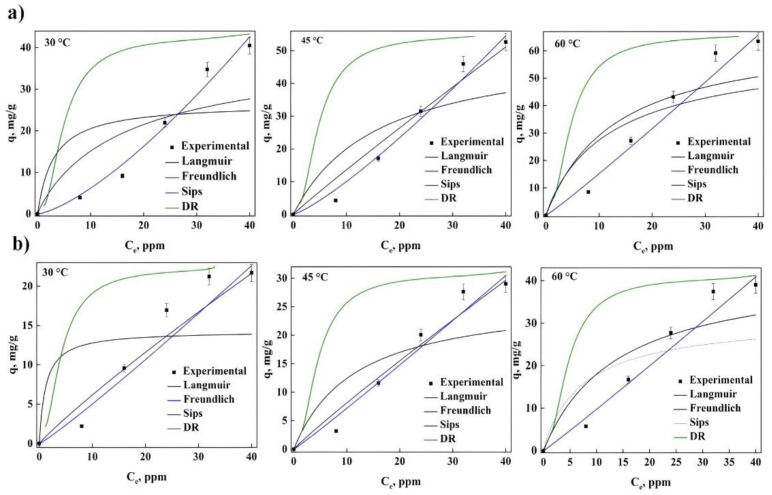
Fitting for the experimental data of the adsorption of GV at different temperatures using different isotherm models: (**a**) WLW and (**b**) NWL.

**Figure 5 molecules-27-06442-f005:**
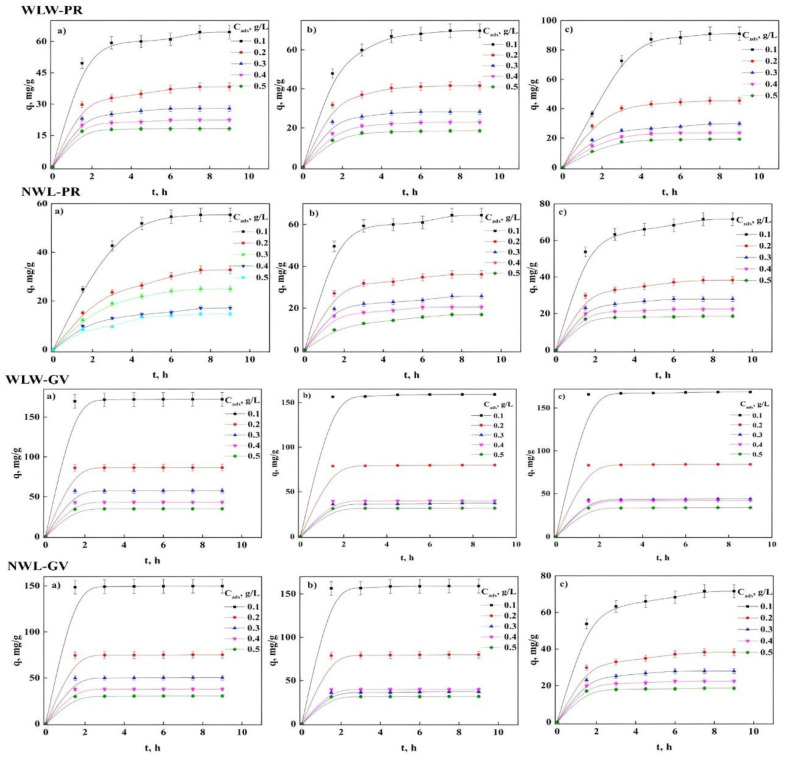
Contact time of the adsorption process of PR and GV in WL: (**a**) 30, (**b**) 45, and (**c**) 60 °C.

**Figure 6 molecules-27-06442-f006:**
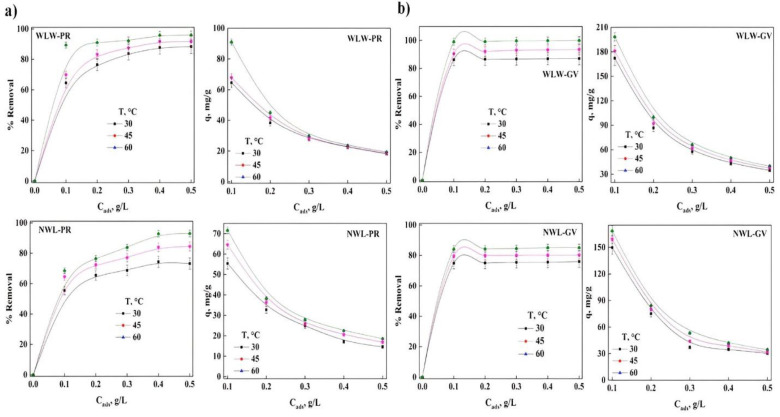
Effect of adsorbent concentration on removal percentage and adsorption capacity: (**a**) PR and (**b**) GV.

**Figure 7 molecules-27-06442-f007:**
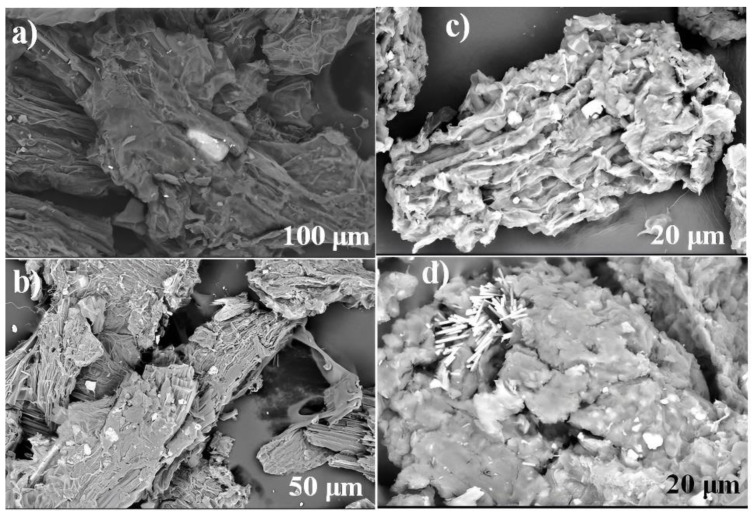
SEM micrographs of WLW after adsorption: for PR (**a**,**b**); for GV (**c**,**d**).

**Figure 8 molecules-27-06442-f008:**
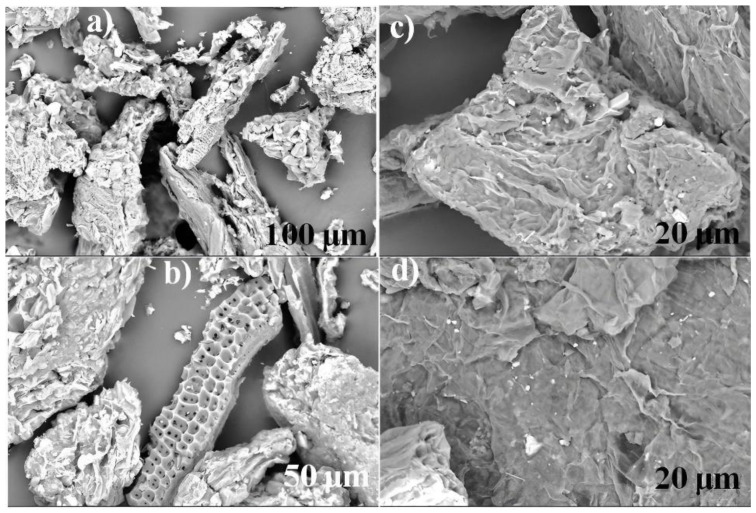
SEM micrographs of NWL after adsorption: for PR (**a**,**b**); for GV (**c**,**d**).

**Figure 9 molecules-27-06442-f009:**
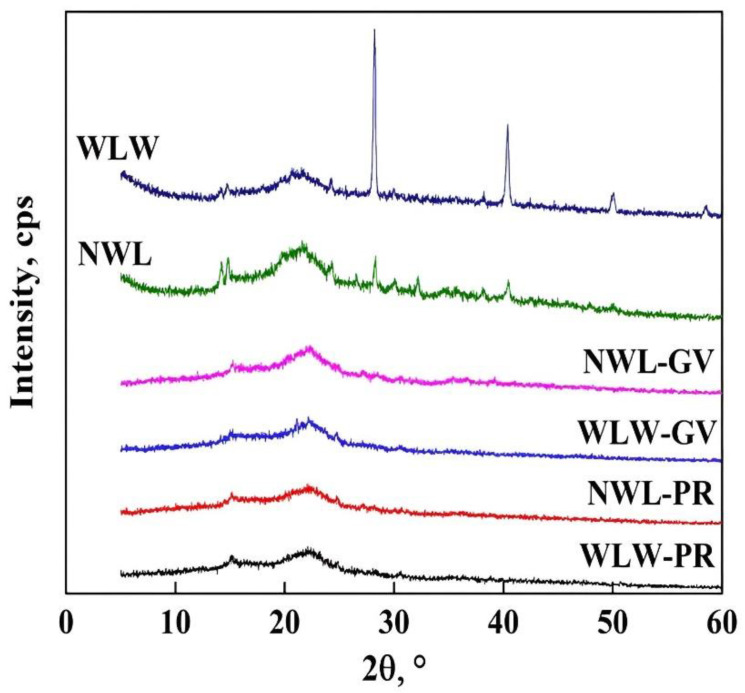
XRD patterns of WL with different pretreatments.

**Figure 10 molecules-27-06442-f010:**
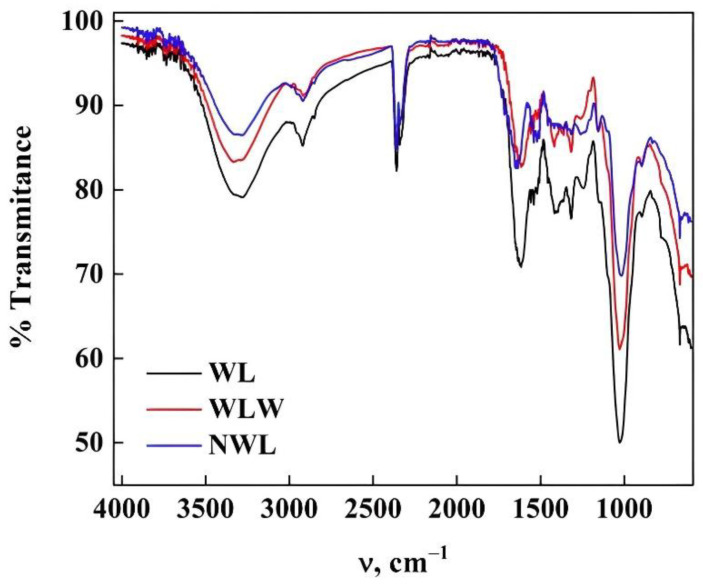
ATR-FTIR spectra of WL with the different treatments.

**Figure 11 molecules-27-06442-f011:**
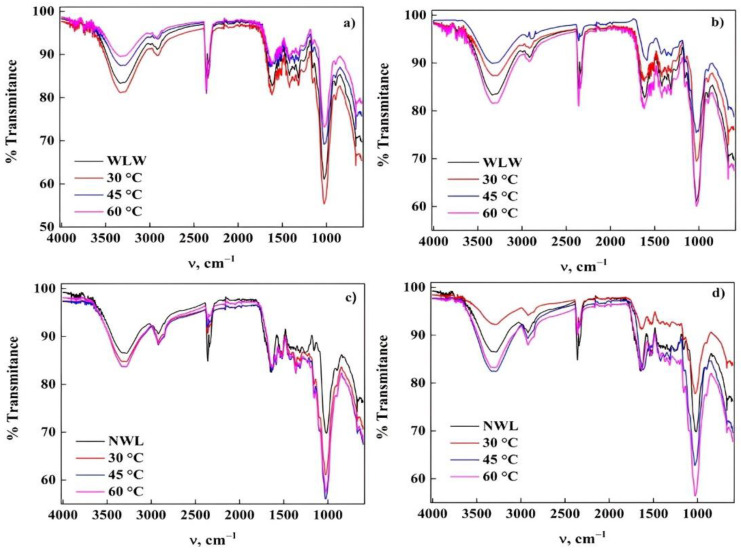
ATR-FTIR spectra of WL after adsorption of dyes: (**a**) WLW-PR; (**b**) WLW-GV; (**c**) NWL-PR and (**d**) NWL-GV.

**Table 1 molecules-27-06442-t001:** Non-linear adsorption isotherm models [36].

Model	Equation	
Langmuir	$q_{e}=\frac{q_{m}K_{L}C_{e}}{1+K_{L}C_{e}}$	q_e_ (mg/g), equilibrium adsorption capacity. Ce (mg/L) is the equilibrium concentration of the dye in the liquid, V (L) is the volume of the dye solution, and m (g), is the mass of the adsorbent. q_m_, is the maximum adsorbed capacity (mg/g). K_L_ (L/mg), Langmuir equilibrium constant. K_F_ ((mg/g)(L/mg)1/n, Freundlich constant indicating the adsorption capacity of the adsorbent, n (dimensionless), is the exponent of the Freundlich model indicates the intensity of adsorption. k_DR_ (mol/J)^2^ constant related to the average energy of adsorption, ε (J/mol), Poliani potential. K_s_ (L/mg) Sips equilibrium constant, β (dimensionless) is the exponent of the Sips model related to the heterogeneity of the system
Freundlich	$q_{e}=K_{F}C_{e}^{\frac{1}{n}}$	
Dubinin-Radushkevich (DR)	$q_{e}=q_{m}\exp\left(-k_{\mathrm{DR}}\varepsilon^{2}\right)$	
Sips	$q_{e}=\frac{q_{m}{\left(K_{s}C_{e}\right)}^{\beta }}{1+{\left(K_{s}C_{e}\right)}^{\beta }}$	

**Table 2 molecules-27-06442-t002:** Parameters for the isotherm models and PR adsorption with WL.

Model	WLW			NWL		
	30 °C	45 °C	60 °C	30 °C	45 °C	60 °C
**Langmuir** K_L_ (L/mg) q_m_ (mg/g) R_L_ R^2^ Δq, %	0.761 14.22 0.55–0.78 0.591 31.681	0.196 24.95 0.25–0.51 0.772 3.951	0.054 34.57 0.08–0.1 0.852 15.676	0.621 21.63 0.43–0.69 0.717 6.235	0.218 21.24 0.23–0.47 0.753 5.412	0.089 24.11 0.09–0.14 0.812 6.578
**Freudlich** K_F_ (mg/g) (L/mg)^1/n^ n R^2^ Δq, %	1.423 1.052 0.922 1.389	1.893 1.088 0.958 0.415	1.928 1.026 0.982 2.425	1.099 1.048 0.943 3.043	1.434 1.052 0.962 4.936	2.238 1.194 0.975 3.073
**Sips** K_S_ (L/mg) q_m_ (mg/g) n_s_ R^2^ Δq, %	0.098 22.051 2.823 0.978 6.505	0.089 26.415 2.111 0.983 7.509	0.103 37.377 1.973 0.991 21.085	0.128 15.462 3.676 0.998 5.091	0.121 26.764 2.867 0.993 22.023	0.142 22.861 2.792 0.999 5.291
**DR** q_m_ (mg/g) k_DR_ E_a_ (kJ/mol) R^2^ Δq, %	1.787 0.051 3.892 0.936 40.567	2.716 0.046 3.202 0.943 39.351	3.888 0.033 3.105 0.932 37.933	1.801 0.044 3.373 0.908 38.691	2.461 0.039 3.269 0.9222 38.585	3.753 0.031 3.151 0.899 36.425

**Table 3 molecules-27-06442-t003:** Parameters for the isotherm models and GV adsorption with WL.

Models	WLW			NWL		
	30 °C	45 °C	60 °C	30 °C	45 °C	60 °C
**Langmuir** K_L_ (L/mg) q_m_ (mg/g) R_L_ R^2^ Δq, %	0.531 26.672 0.24–0.62 0.726 15.433	0.199 40.362 0.12–0.40 0.747 10.411	0.077 69.991 0.050.20 0.788 4.648	0.840 15.554 0.24–0.62 0.761 12.674	0.246 28.253 0.09–037 0.739 1.156	0.072 45.011 0.03–0.13 0.795 6.652
**Freudlich** K_F_ (mg/g) (L/mg)^1/n^ n R^2^ Δq, %	1.253 0.719 0.982 20.563	0.613 0.819 0.984 2.387	1.392 0.946 0.977 3.759	0.677 0.924 0.949 9.573	0.454 0.958 0.9445 11.805	0.954 0.967 0.977 4.869
**Sips** K_S_ (L/mg) q_m_ (mg/g) n_s_ R^2^ Δq, %	0.968 40.94 0.916 0.688 0.264	0.065 52.09 0.955 0.727 0.434	0.077 65.10 0.932 0.774 1.199	0.088 21.981 0.914 0.796 0.619	0.087 28.985 0.934 0.726 0.031	0.136 38.254 0.942 0.685 0.894
**DR** q_m_ (mg/g) k_DR_ E_a_ (kJ/mol) R^2^ Δq, %	1.956 0.024 4.612 0.969 42.572	3.459 0.029 4.158 0.964 41.784	6.382 0.035 3.591 0.948 16.178	2.409 0.022 4.752 0.967 43.447	2.736 0.025 4.435 0.939 40.508	4.174 0.026 4.363 0.949 35.933

**Table 4 molecules-27-06442-t004:** Thermodynamic parameters of dye adsorption using modified WL.

T, °C	−ΔG, kJ/mol	ΔH, kJ/mol	−ΔS, kJ/mol K	−ΔG, kJ/mol	ΔH, kJ/mol	ΔS, kJ/mol K
WLW-PR				NWL-PR		
30	41.64	83.53	0.051	41.13	98.62	0.646
45	40.14			40.39		
60	38.44			39.82		
WLW-GV				NWL-GV		
30	41.09	35.75	0.224	42.25	67.32	0.127
45	40.43			41.09		
60	39.81			39.81		

**Table 5 molecules-27-06442-t005:** Adsorption kinetic models used for the analysis of experimental data [36,40].

Model	Equation	
Pseudo first order (PFO)	$q=q_{\max}\left[1-\exp\left(-k_{1}t\right)\right]$	q (mg/g), adsorption capacity. C_0_ (mg/L) is the initial concentration of the dye in the liquid, V (L) is the volume of the dye solution and m (g), is the mass of the adsorbent. q_max_ is the maximum adsorbed capacity (mg/g). k_1_ (1/h) is the speed constant of the PPO model. k_2_ (g s/mg) is the speed constant of the PSP model. k_ext_ constant kinetic of Avrami model (h^−1^), n_A_ reflects the changes of the mechanism during the adsorption process. k_Int_ (mg/g h) is the speed constant of the ID model. k_Ext_ (1/h) is the speed constant of the model ED.
Pseudosecond order (PSO)	$q=\frac{t}{\frac{1}{k_{2}{*q}_{\max}^{2}}+\frac{t}{q_{\max}}}$	
Avrami	$q=q_{\max}\left[1-\exp{\left(-k_{A}t\right)}^{n_{A}}\right]$	
Intraparticle diffusion (ID)	$q=k_{\mathrm{ID}}t^{0.5}$	
External diffusion (ED)	$q=\frac{C_{0}V}{m}\left[1-\exp\left(-k_{\mathrm{ext}}*t\right)\right]$	

**Table 6 molecules-27-06442-t006:** Kinetic parameters for PR adsorption with WLW.

Model	0.1 g/L			0.3 g/L			0.5 g/L		
	30 °C	45 °C	60 °C	30 °C	45 °C	60 °C	30 °C	45 °C	60 °C
**PFO** q_max_, mg/g k_1_, h^−1^ R^2^ Δq, %	45.60 1.017 0.996 1.131	62.71 0.742 0.998 0.259	97.85 0.332 0.998 2.076	27.64 1.139 0.995 0.693	27.95 1.118 0.997 0.431	29.20 0.656 0.995 0.846	18.18 1.861 0.998 0.168	18.48 0.904 0.999 0.121	19.59 0.612 0.994 0.773
**PSO** q_max_, mg/g k_2_, h^−1^ R^2^ Δq, %	59.803 8 × 10^7^ 0.953 2.963	63.72 1 × 10^8^ 0.904 3.546	73.78 1 × 10^6^ 0.675 7.723	26.46 3 × 10^6^ 0.968 2.160	26.88 4 × 10^6^ 0.967 1.969	26.31 5 × 10^5^ 0.874 4.791	17.97 6 × 10^7^ 0.996 0.633	17.41 1 × 10^7^ 0.936 2.478	17.40 3 × 10^7^ 0.831 3.875
**Avrami** q_max_, mg/g k_A_, h^−1^ n_A_ R^2^ Δq, %	62.69 1.307 0.781 0.996 1.333	69.33 0.961 0.771 0.998 0.587	97.85 0.571 0.583 0.998 3.654	27.46 1.339 0.851 0.995 1.548	27.95 1.288 0.868 0.997 0.982	29.19 0.839 0.781 0.995 0.999	18.18 1.967 0.930 0.999 0.587	18.47 0.997 0.906 0.999 1.256	19.59 0.756 0.811 0.994 1.587
**ID** k_id_, mg/g h ^0.5^ R^2^ Δq, %	25.72 0.776 3.875	27.67 0.841 7.738	33.21 0.976 3.875	11.34 0.729 9.121	11.52 0.723 8.896	11.49 0.883 0.392	7.612 0.591 10.25	7.500 0.863 7.686	7.616 0.863 9.470
**ED** k_ext_, h^−1^ R^2^ Δq, %	78.78 0.339 22.45	32.38 0.146 17.68	1.917 0.000 4.036	0.216 0.586 0.999	0.226 0.596 1.075	0.212 0.801 1.679	0.101 0.129 14.27	0.098 0.408 14.95	0.101 0.613 15.48

**Table 7 molecules-27-06442-t007:** Kinetic parameters for PR adsorption with NWL.

Model	0.1 g/L			0.3 g/L			0.5 g/L		
	30 °C	45 °C	60 °C	30 °C	45 °C	60 °C	30 °C	45 °C	60 °C
**PFO** q_max_, mg/g k_1_, h^−1^ R^2^ Δq, %	28.15 0.450 0.998 1.424	39.90 0.469 0.994 0.471	51.24 0.847 0.995 0.850	9.673 1.081 0.998 0.670	18.46 1.122 0.990 1.677	23.42 1.117 0.997 1.008	5.865 2.015 0.978 2.743	12.80 1.222 0.987 1.210	16.63 1.273 0.991 1.737
**PSO** q_max_, mg/g k_2_, h^−1^ R^2^ Δq, %	23.35 3 × 10^8^ 0.762 5.780	33.55 2 × 10^7^ 0.797 6.097	47.94 9 × 10^5^ 0.929 3.423	9.277 1 × 10^6^ 0.963 2.313	17.77 6 × 10^6^ 0.961 3.155	22.52 6 × 10^6^ 0.966 2.530	5.815 1 × 10^6^ 0.976 3.067	13.71 5 × 10^6^ 0.995 1.616	16.15 1 × 10^7^ 0.964 2.082
**Avrami** q_max_, mg/g k_A_, h^−1^ n_A_ R^2^ Δq, %	58.31 0.578 0.731 0.994 2.587	62.69 1.063 0.956 0.999 1.587	69.53 0.998 0.931 0.995 2.025	25.69 0.589 0.740 0.999 1.871	24.47 1.007 0.978 0.985 2.987	27.46 1.142 0.997 0.999 1.987	15.05 0.589 0.728 0.978 3.658	16.77 0.640 0.777 0.991 2.587	18.35 1.746 1.000 0.999 2.698
**ID** k_id_, mg/g h^0.5^ R^2^ Δq, %	10.36 0.942 5.831	14.88 0.951 5.390	20.74 0.816 7.716	3.979 0.734 8.735	7.627 0.744 7.692	9.660 0.735 8.451	2.476 0.647 7.407	5.316 0.730 8.529	6.916 0.716 7.936
**ED** k_ext_, h^−1^ R^2^ Δq, %	0.172 0.871 77.10	0.413 0.991 60.16	1.802 0.822 37.18	0.042 0.169 2.806	0.101 0.364 1.264	0.154 0.471 1.343	0.024 0.000 15.37	0.060 0.227 17.87	0.087 0.281 8.603

**Table 8 molecules-27-06442-t008:** Kinetic parameters for GV adsorption with WLW.

Model	0.1 g/L			0.3 g/L			0.5 g/L		
	30 °C	45 °C	60 °C	30 °C	45 °C	60 °C	30 °C	45 °C	60 °C
**PFO** q_max_, mg/g k_1_, h^−1^ R^2^ Δq, %	172.2 2.835 1.000 0.04	180.2 1.906 0.999 0.181	196.2 1.742 0.999 0.422	57.47 3.885 1.000 0.079	61.73 2.488 0.999 0.214	65.92 1.996 0.999 0.175	34.76 2.861 1.000 0.046	37.20 3.464 0.999 0.240	39.81 3.095 0.999 0.223
**PSO** q_max_, mg/g k_2_, h^−1^ R^2^ Δq, %	171.7 3 × 10^6^ 0.998 0.140	178.4 4 × 10^6^ 0.997 0.593	193.63 5 × 10^6^ 0.994 0.973	57.44 0.640 0.999 0.099	62.37 0.298 0.998 0.208	67.13 0.146 0.998 0.575	34.68 6 × 10^6^ 0.999 0.139	37.16 4 × 10^6^ 0.999 0.278	39.75 3 × 10^6^ 0.999 0.289
**Avrami** q_max_, mg/g k_A_, h^−1^ n_A_ R^2^ Δq, %	172.2 2.835 1.000 1.000 0.987	180.2 2.528 0.754 0.999 2.584	196.2 2.165 0.805 0.999 0.898	57.47 4.620 0.841 1.000 0.587	61.73 3.056 0.814 0.999 0.483	65.92 2.230 0.895 0.999 0.258	34.76 2.998 0.954 1.000 0.870	37.21 3.578 0.968 0.999 0.560	39.81 3.174 0.975 0.999 0.287
**ID** k_id_, mg/g h^0.5^ R^2^ Δq, %	72.42 0.572 10.64	75.53 0.586 10.28	82.12 0.607 9.882	24.20 0.525 10.67	26.96 0.552 10.41	27.64 0.575 10.31	14.62 0.634 10.64	15.67 0.535 10.47	16.77 0.538 10.67
**ED** k_ext_, h^−1^ R^2^ Δq, %	3 × 10^6^ 0.000 0.337	1 × 10^6^ 0.000 4.280	2 × 10^6^ 0.000 6.555	1 × 10^6^ 0.472 6.435	5 × 10^8^ 0.273 3.027	2 × 10^7^ 0.081 0.282	0.843 0.801 6.062	1.274 0.921 2.805	1.879 0.988 0.044

**Table 9 molecules-27-06442-t009:** Kinetic parameters for GV adsorption with NWL.

Model	0.1 g/L			0.3 g/L			0.5 g/L		
	30 °C	45 °C	60 °C	30 °C	45 °C	60 °C	30 °C	45 °C	60 °C
**PFO** q_max_, mg/g k_1_, h^−1^ R^2^ Δq, %	149.5 3.410 1.000 0.055	158.5 2.874 0.999 0.147	167.9 2.916 0.999 0.144	36.83 2.655 0.999 0.478	43.61 2.714 0.999 0.381	50.18 3.128 0.999 0.119	30.29 2.761 0.999 0.129	31.64 2.946 1.000 0.109	33.51 3.083 1.000 0.016
**PSO** q_max_, mg/g k_2_, h^−1^ R^2^ Δq, %	149.3 2 × 10^6^ 0.999 0.110	158.1 3 × 10^6^ 0.999 0.239	167.63 3 × 10^6^ 0.999 0.231	36.71 4 × 10^6^ 0.999 0.605	43.48 1 × 10^6^ 0.999 0.498	50.10 1 × 10^6^ 0.999 0.498	30.21 1 × 10^7^ 0.999 0.239	31.56 9 × 10^6^ 0.999 0.192	33.45 7 × 10^6^ 0.999 0.039
**Avrami** q_max_, mg/g k_A_, h^−1^ n_A_ R^2^ Δq, %	149.5 3.801 0.897 1.000 0.874	158.5 3.483 0.825 0.999 0.501	168.0 3.587 0.813 0.999 1.005	74.95 3.514 0.967 1.000 0.981	36.83 3.026 0.877 0.999 0.387	43.60 3.080 0.881 0.999 0.541	30.29 2.790 0.989 0.999 0.365	31.64 3.052 0.965 1.000 0.248	33.50 3.202 0.963 0.999 0.111
**ID** k_id_, mg/g h^0.5^ R^2^ Δq, %	62.93 0.527 10.70	66.73 0.541 10.55	70.70 0.538 10.54	15.51 0.558 10.15	18.36 0.553 10.26	21.12 0.533 10.58	12.75 0.541 10.54	13.31 0.536 10.58	14.11 0.539 10.49
**ED** k_ext_, h^−1^ R^2^ Δq, %	2 × 10^6^ 0.000 13.73	4 × 10^6^ 0.000 10.50	1 × 10^6^ 0.000 7.607	285.5 0.987 21.01	113.6 0.908 32.20	92.11 0.802 13.64	0.340 0.518 10.39	0.434 0.594 9.608	0.653 0.720 7.522

**Table 10 molecules-27-06442-t010:** Comparison of the adsorption capacity of different bioadsorbents.

Adsorbent	q, mg/g	Dyes	Reference
*Nymphaea alba*	9.66	Phenol Red	[3]
*Diplazium esculentum stems*	44.88	Gentian Violet	[12]
Water Lily roots	42.65	Methylene Blue	[20]
Raw corn stalk biochar	325.1 387.7	Methylene Blue Gentian Violet	[24]
*Eichhornia crassipes*/ZnO NP	40 39.9	C. I: Blue Red 43 C. I. Basic Blue 3	[27]
*Metroxylon sagu*	238.82	Methylene Blue	[31]
Sulfanated carbon from *Eichhornia crassipes*	19.5	Methylene Blue	[32]
*Eichhornia crassipes*/NC	20.83	Methylene Blue	[34]
*Eichhornia crassipes* untreatment	98.8 87.7 44.4 62.8	Methylene Blue Crystal Violet Phenol Red Methyl Orange	[35]
*Eichhornia crassipes* roots	9.95	Methylene Blue	[37]
*Eichhornia crassipes* untreatment	161.64	Methylene Blue	[38]
*Hordeum vulgare* L.	3.50	Phenol Red	[42]
*Eichhornia crassipes* treated with water	200.5 193.5	Methylene Blue Methyl Orange	[49]
*Eichhornia crassipes* treated with NaOH	226.5 158.5	Methylene Blue Methyl Orange	[49]
*Eichhornia crassipes* treated with water	73.78 196.1	Phenol Red Gentian Violet	This study
*Eichhornia crassipes* treated with NaOH	47.94 167.9	Phenol Red Gentian Violet	

**Table 11 molecules-27-06442-t011:** Elemental analysis of WL treated with water and NaOH.

Bioadsorbent	wt.%				
	C	O	Al	Si	Ca
WLW	59.64	38.05	0.20	0.72	2.08
NWL	62.16	26.49	0.18	0.52	0.71
WLW-PR	56.67	39.33	0.44	1.47	2.16
NWL-PR	48.51	43.21	0.79	2.17	0.28
WLW-GV	48.35	45.95	0.30	3.05	1.64
NWL-GV	50.61	45.99	1.01	0.52	1.24

**Table 12 molecules-27-06442-t012:** Network parameters of the different biomaterials before and after the adsorption process.

Biomaterials	a_0_, nm
WLW	0.829
NWL	0.824
WLW-PR	0.856
NLW-PR	0.815
WLW-GV	0.849
NWL-GV	0.813

## Data Availability

Not applicable.

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
