# Peer review of "Removal of Anionic and Cationic Dyes Present in Solution Using Biomass of Eichhornia crassipes as Bioadsorbent"

_molecules, 2022, doi:10.3390/molecules27196442_

Round 1
Reviewer 1 Report
A study of the capacity of Eichhornia crassipes biomass to absorb two industrial dyes was carried out. Factors such as: biomass treatment, initial concentration of the pollutant, temperature, biomass concentration (absorbent) are analyzed. Kinetic and thermodynamic models are presented.
I recommend acceptance of the manuscript, after major corrections.
1. There are no citations to support the information presented in lines 33-35.
2. There are no citations to support the information presented in lines 36-38.
3. Line 39. What are the adverse public health effects?
4. In general, the referencing of the information presented in the Introduction should be restructured. It should be cited paragraph by paragraph, in order to identify the articles/books that support the information presented.
5. What is the meaning of the acronyms PR and SIPS?
6. Section 2.1 (xxx) does not report the effect of initial concentration; the effect of temperature is reported. The resolution of Figure 2 should be improved (it is recommended to edit the images with LaTeX).
7. Fig. 2 reports the effect of temperature, but does not report the effect of initial concentration.
8. What is the meaning of NWL?
9. Describe each one of the variables of the Non-linear adsorption isotherm models.
10. What is the analytical technique used for the quantification of each of the dyes? (Mention it in the abstract)
11. it is advisable to indicate in the title that Eichhornia crassipes biomass is used.
12. The resolution of all graphs should be improved.
Author Response
Dear editor, many thanks for your comments and observations.
1 y 2. The observation is considered by placing the respective references in each of the paragraphs of the manuscript that required it.
- The observation is considered by placing the respective references in each of the paragraphs of the manuscript that required it.
- The observation is considered by placing the respective references in each of the paragraphs of the manuscript that required it and rewriting some parts of the introduction.
- Line 50 mentions that Phenol Red (PR) and Sips is the last name of the creator of the model (Robert Sips 1948)
- The graph was improved and the discussion of the results of the effect of the initial concentration of the dyes concerning the percentage of removal was added.
- The discussion of the results of the effect of the initial concentration of the dyes concerning the percentage of removal was added to the manuscript
- The reference of WLW (Water Lily treatment with water) and NWL (Water Lily treatment with NaOH) was considered in the introduction.
- Table 1 describes each of the variables of the adsorption isotherm models.
- In the abstract it was mentioned that UV-Vis spectroscopy was used to determine the residual concentration of the dyes
- The title of the manuscript was changed to take into account the reviewer's suggestion that we believe is more appropriate
- Graphs 2-6 and 9-11 have improved resolution
Reviewer 2 Report
The topic is very interesting and the article is very well represented. However, minor revisions are still required.
1- The following articles should be cited
https://www.mdpi.com/2073-4360/14/13/2732
https://www.researchgate.net/publication/267791508_Biosorption_of_cationic_dyes_from_aqueous_solution_by_water_hyacinth_roots
https://www.sciencedirect.com/science/article/pii/S2215153221001446
2- Aim of the work is very clear and precise. Thanks
3- Each instrument should be described with town and country
4- A comparative table should be provided to evaluate the efficiency of water lily (Eichhornia crassipes) as bioadsorbent against other natural bioadsorbents e.g. Potamogeton lucens, 86
5- Salvinia hergozi, Myriophyllum spicatum, Nymphaea alba, Cabomba sp. For removal of organic dyes.
6- Future research plan should be provided
7- Conclusion should be more precise
Best wishes
Author Response
Dear editor, many thanks for your comments and observations.
- References were analyzed and added to the manuscript.
- Improved the objective of the works so that it is clearer
3, Brand, state and country of each of the instruments used in the study were added
- At this point, table 10 was added to better visualize the adsorption capacity of these adsorbents.
- At this point, table 10 was added to better visualize the adsorption capacity of these adsorbents.
- This paragraph was added at the end of the conclusion
- The conclusion was restructured in such a way that only the most relevant is mentioned
Reviewer 3 Report
Comments to the authors
The manuscript reports interesting results concerning the utilization of lily bioadsorbent for the removal of gentian violet and phenol red dyes. Preparation of an economical and efficient adsorbent for the decontamination of organic dyes have received increased attention in the last few decades. The present work affords a promising strategy for the designing economical and effective adsorbent for the removal of gentian violet and phenol red dyes. The manuscript contain huge grammatical mistakes and lacks novelty and therefore cannot be accepted in the present form. I suggest major revision for quality enhancement, which are given bellow.
11. The manuscript contains huge grammatical and typing mistakes which should be carefully read and must be removed and the language need sufficient improvement.
22. Line 22-23, the sentence “The thermodynamic analysis for both dyes showed at the spontaneous and endothermic process” is senseless. Line 37, what does “generates” means here? Such types of mistakes are present in the whole manuscript. Correct it.
33. Line 55-57, separate sentence must be used for the toxicity of phenol red.
44. Line 61-62, provide references to each treatment methods.
55. Line 80 and 81 replace the word “compounds” with “materials”.
66. Why some references are repeated for several sentences.
77. Line 99, full stop after the word dye must be removed.
88. Line 101-108, split this single sentence to 2 or 3 sentences. Also correct its grammar.
99. The author mentioned in the experimental section that 81.5 g of a WL powder is obtained. Mention the quantity of the starting WL and also mention the %age of weight loosed.
110. Carefully check all the equations used in the whole manuscript for any error.
111. Line 496 and 496, what is MB and MO? This section look like copied from other paper.
112. The authors claimed that WL was modified with water and NaOH. How water can modify the WL? Water can be used for washing not for any modification.
113. Line 534-540 in conclusion, split this single sentence to several sentences and clarify the meaning.
114. The quality of the adsorption study figures are very week. Improve their quality.
115. Correct the figure caption of the last figure as Figure 11.
116. The results and discussion section need sufficient attention due to large grammatical mistakes, e.g. line 114-116 the word “with” is used three times in a single sentence.
117. WLW and NLW stands for what?
118. Discuss the toxicity of dyes in the introduction using these article, Water 2022, 14, 242 and J Env Chem Eng 8 (2020) 104364
119. The adsorption section need major revision. This discussion of this section is too lengthy which need to be shorten.
220. The FTIR discussion need major revision due to grammatical mistakes.
221. The authors should properly correlate the FTIR data with the results presented in the manuscript.
Author Response
Dear editor, many thanks for your comments and observations.
- The manuscript was revised and both grammatical and typographical errors were corrected, taking care to improve the language of the work.
- The manuscript was thoroughly revised to improve its understanding.
- This sentence was separated to obtain two paragraphs
- References are echoed in almost all methods, so there is no point in doing this fix as it would be too redundant
- Replaced the suggested word not only in this part of the manuscript but also where it was appropriate to do so
- The regencies were taken into account several times because they have the same idea or referring to the sentence where they were cited
- State typo was removed
- The sentence was divided into two sentences, in addition to correcting grammar
- The data requested by the reviewer were added to the manuscript.
- The equations were reviewed together with the cited references to avoid errors in the manuscript.
- In this paragraph, an error was made, since they are different dyes but with a similar experimental methodology
- With the increase in temperature in the water, the colour and some superficial components that interfere with adsorption were eliminated, since tests were started with the untreated bioadsorbent and no appreciable adsorption capacity was observed.
113 The conclusion was divided into several sentences to clarify the central ideas of each of them.
- All figures have been increased in quality and resolution
- Fixed figure 11 footer
- The manuscript was modified to avoid as much as possible the excessive use of this type of connectors
- These acronyms were defined in the introduction and experimentation section
- The work of Khan et al (2022) was consulted and considered for this manuscript.
- This part of the manuscript was restructured in an attempt to be clearer and more precise in the discussion of the results and the comparison with what is reported in the literature.
- Grammatical and typographical errors were corrected, in addition to being clearer in the discussion of the FTIR results.
- The results in characterization (SEM-EDS, XRD, pHpzc and FTIR) were related to the results obtained in the adsorption of the appropriate dyes.
Round 2
Reviewer 1 Report
The authors made the suggested corrections
Author Response
Dear reviewer, thank you very much for all your contributions to improve the manuscript.
Reviewer 3 Report
The manuscript is sufficiently improved however the language and figure quality need further improvement.
Author Response
Dear reviewer, thank you very much for your comments and contributions toimprove the quality of the manuscript. He told him that the resolution of
the graphs and their legends were improved and the error bars were added.
The language was revised again in order to have a better quality of the
manuscript.